# Self-Enrichment in Globular Clusters: The Crucial Role Played by Oxygen

**Erendira Huerta-Martinez [1,2], Claudio Gavetti [3] and Paolo Ventura [1,*]**

1    INAF, Osservatorio Astronomico di Roma, Via Frascati 33, Monte Porzio Catone, 00078 Rome Italy
2    Academia de Fisica, Universidad Autónoma de la Ciudad de México, Casa Libertad. Ermita Iztapalapa 4163 Alcaldía Iztapalapa, Ciudad de México C.P. 09620, Mexico
3    Dipartimento di Matematica e Fisica, Università degli Studi Roma Tre, Via della Vasca Navale 84, 00100 Rome, Italy
\*    Correspondence: paolo.ventura@inaf.it

**Abstract:** Results from photometry and high-resolution spectroscopy have demonstrated the existence of multiple populations in globular clusters, where one or more stellar generations of stars coexist with the original population. We study the possibility that the new generation(s) of stars formed from the gas lost by intermediate mass (4–8 $M_\odot$) stars during the asymptotic giant branch (AGB) evolution, possibly after dilution with residual pristine gas in the cluster. To this aim, we confront the chemistry of the AGB ejecta with the distribution of the chemical composition of stars in NGC 6402. We find that a satisfactory consistency between the observational evidence and the self-enrichment by AGBs hypothesis is reached if the mass-loss rates experienced for the latter stars is ∼10 times smaller than found for the solar metallicity counterparts of the same mass. We also comment on the importance of the knowledge of the oxygen abundance as a key indicator of the extent of the nucleosynthesis at which the gas from which the stars belonging to the second generation of the clusters formed, and the degree of dilution with pristine gas.

**Keywords:** stars: AGB and post-AGB; stars: evolution; stars: abundances





## 1. Introduction

Globular clusters (GC) have been historically considered as the best example of simple stellar populations, made up of coeval stars sharing the same chemical composition. This assumption allowed to test stellar evolution theories and to deduce the age and the distance of most of the Galactic GC [1]. This simple understanding has been challenged by several observational evidences collected in the last decades, which suggest the coexistence of different stellar populations in the same cluster. Results from high-resolution spectroscopy outlined clear star-to-star differences in the surface chemical composition, which define chemical patterns involving the light elements, from carbon to aluminium; these findings first regarded red giant branch (RGB) stars [2], then were extended to unevolved stars of the MS and SGB [3–5]. An important breakthrough in this context came with the discoveries of the multiple populations in $\omega$ Centauri [6–8], and of the multiple main sequences in the GC NGC2808 [9,10]; later studies confirmed the presence of multiple main sequences and giant branches in several Galactic GC [11]. An additional push towards the need of a revision of the paradigm that GC are composed by a single stellar population came from the studies of the HB morphology of some GC, which can be interpreted only by invoking the presence of two or more stellar components, formed with different helium mass fractions [12–14].

This impressive body of observational evidences suggested the action of a self-enrichment mechanism in GC, according to which further generations of stars formed within GC, from the ashes of stars belonging to the first generation. The most popular pollutants proposed as responsible for such a self-enrichment mechanism are fast-rotating massive stars [15], massive asymptotic giant branch (AGB) stars [16], massive binaries [17] and

supermassive stars [18]. An exhaustive discussion of the critical points related to all the proposed scenarios is given in [19].

In the present work we focus on the self-enrichment by AGBs scenario, and discuss whether the abundance patterns traced by the stars in Galactic GC can be explained on the basis of the chemical yields of the latest generation of massive AGB stellar models. Previous explorations on this argument relied on the straight comparison between the mass fraction of the chemical species of the stars within a given GC exhibiting the most anomalous chemical composition, and the chemistry of the ejecta of AGB stars with the same metallicity of the cluster stars belonging to the original population. In the present analysis we make a step forward, in that we also take into account the possibility that the formation of the new stellar generations took place after the AGB ejecta mixed with some residual pristine gas in the cluster: this is now made possible by the recent determination of the helium spread among the stars belonging to most of the clusters investigated [20], a quantity that is tightly connected to the degree of dilution.

Part of this investigation is dedicated to discuss the central role played by the distribution of the oxygen abundances of the stars in a given cluster, which allows the possibility of testing the hypothesis that the second generation of stars formed from the ashes of AGB stars belonging to the original stellar population of the cluster: this is because, among the different chemical species, the oxygen content of the ejecta of massive AGBs is the most sensitive to the metallicity of the stars considered and to the input physics adopted to build the stellar models, particularly the description of the mass-loss mechanism.

The paper is structured as follows: the evolution code used to model stellar evolution is described in Section 2; the most relevant properties of AGB stars are treated in Section 3, whereas Section 4 is dedicated to the presentation of the self-enrichment scenario by massive AGBs in GC; in Section 5 we test the self-enrichment by AGBs possibility against the observations of stars in the cluster NGC 6402; in Section 6 we explore the information that can be obtained by the observed distribution of the abundances of the various chemical species in a goven cluster, outlining the importance of oxygen. Finally, the conclusions are given in Section 7.

## 2. The ATON Code for Stellar Evolution

The evolutionary sequences used in the present investigation were calculated by means of the ATON code for stellar evolution, which was described in detail in [21], where the interested reader can find an accurate analysis of the numerical structure and of the input physics adopted. Here we recall the physical ingredients most relevant for the description of the AGB phase.

Nuclear burning and mixing of chemicals are self-consistently coupled by means of a diffusive scheme, according to the prescription by [22]. The temperature gradient with regions unstable to convective motions is found via the full spectrum of the turbulence (FST) model [23]. The rate of mass loss during the phases when the star is oxygen-rich at the surface is modelled via the treatment of [24]. The free parameter entering the [24]'s recipe is set to $\zeta = 0.02$, based on the calibration of the luminosity function of lithium-rich stars in the Magellanic Clouds by [25]. For the carbon star phases we used the results on mass loss published from the Berlin group [26,27]. The surface molecular opacities were calculated via the AESOPUS tool [28], which allows considering the effects of the variation of the surface abundances of the CNO species.

We will refer to the evolutionary sequences calculated with the input given above as "standard". Some further sequences calculated with different values of the parameter $\zeta$ will be also considered, with the aim of investigating the role of the mass loss treatment on the description of AGB stars and on the chemical composition of their ejecta.

## 3. The Evolution of Massive AGB Stars

The evolution through the asymptotic giant branch is experienced after the end of core helium burning by all the stars with mass in the 1–8 $M_{\odot}$ range [29–31]. The AGB

phase is extremely important to understand the pollution expected from the stars, because it is during the AGB evolution that most of the mass of the envelope is lost via stellar winds, which are dispersed into the interstellar medium. The energy supply of AGB stars provided for most of the time by a H-burning shell; the only exceptions are the periodic ignitions of a helium-rich buffer laying above the degenerate core, which are commonly referred to as thermal pulses (TP), because helium burning is started in conditions of thermal instability [32]. At each TP ignition H-burning is temporarily extinguished, and is restored later, after the contraction of the surface regions of the star.

### 3.1. The Second Dredge-Up

The AGB phase of $M \geq 4\,M_\odot$ stars is characterized by a convective episode taking place after the core helium burning phase, during which the surface convection sinks inwards, until reaching layers previously touched by CNO nucleosynthesis; this event is known as second dredge-up (SDU), to distinguish it from the first convective event in the life of stars, which happens during the ascending of the RGB. The main outcome of the SDU is the helium enrichment of the surface regions: the helium mass fraction increases, until reaching $Y \sim 0.35$ in $M > 5\,M_\odot$ stars. This result is rather robust, as it is not affected by all the uncertainties in the modelling of the TP phase [33], and is fairly independent of the metallicity of the star [34].

### 3.2. Hot Bottom Burning

With the terminology "massive AGBs" we refer to the stars of initial mass $M \geq 4\,M_\odot$, which during the AGB evolution experience the physical mechanism known as hot bottom burning (HBB), originally described in [35]. HBB consists in the activation of proton-capture nucleosynthesis in the innermost zones of the convective envelope, once the temperatures in those regions exceeds $\sim 30$ MK [36]. The activation of HBB is favoured by the steep profiles of the thermodynamic variables in the regions between the border of the degenerate core and the base of the external mantle, which triggers a partial overlapping of the convective envelope with the CNO burning shell. These physical conditions are achieved when the core mass of the star is above $\sim 0.8\,M_\odot$ [37], which reflects into initial masses above $\sim 4\,M_\odot$.

The occurrence of HBB affects significantly the evolution through the AGB. On the physical side, the ignition of HBB triggers a fast increase in the luminosity of the star [35,36], which becomes significantly higher than expected on the basis of the classic core mass-luminosity relation by [38]. This reflects into shorter evolutionary times and higher mass-loss rates experienced [39]. On the chemical side, the rapidity of convective currents favour a fast mixing within the whole envelope, thus the byproducts of the nuclear activity at the base of the external mantle are carried to the outer regions of the star: the surface chemistry changes according to the equilibria among the various species established at the base of the envelope.

Figure 1 shows the variation of the main physical quantities characterizing the AGB evolution of a $6\,M_\odot$ model star of metallicity $Z = 10^{-3}$. Figure 2 shows the variation of the surface abundances of the most stable isotopes taking part to CNO cycling (top panels), and of magnesium and aluminium (bottom panels), as a function of the AGB time (counted since the beginning of the TP phase) and of the mass of the star. The latter choice allows to better understand the expected chemical composition of the ejecta.

In Figure 1 it is possible to see the fast increase in the overall luminosity (left panel) occurring at the ignition of HBB, which stars $\sim 3$ Kyr after the start of the AGB phase. In this specific case the luminosity grows until reaching $\sim 8 \times 10^4\,L_\odot$, then decreases down to $\sim 3 \times 10^4\,L_\odot$ towards the end of the AGB lifetime. The decrease in the luminosity during the second part of the AGB evolution is due to the gradual loss of the external mantle, which triggers first a gradual decrease and then the turning off of HBB [25]. The temperature at the bottom of the envelope ($T_{bce}$, right panel) also increases during the initial part of the AGB phase, until reaching $\sim 110$ MK; the loss of the envelope favours a general cooling of the outer regions of the star, so that $T_{bce}$ drops to $\sim 70$ MK towards the final part of

this evolutionary phase. The mass-loss rate is seen in the right panel of Figure 1 to follow the same trend of luminosity: therefore, most of the mass of the star is lost during the evolutionary phases around the luminosity peak, as also clear in the run of the stellar mass with time, reported in the left panel of Figure 1 (red line).

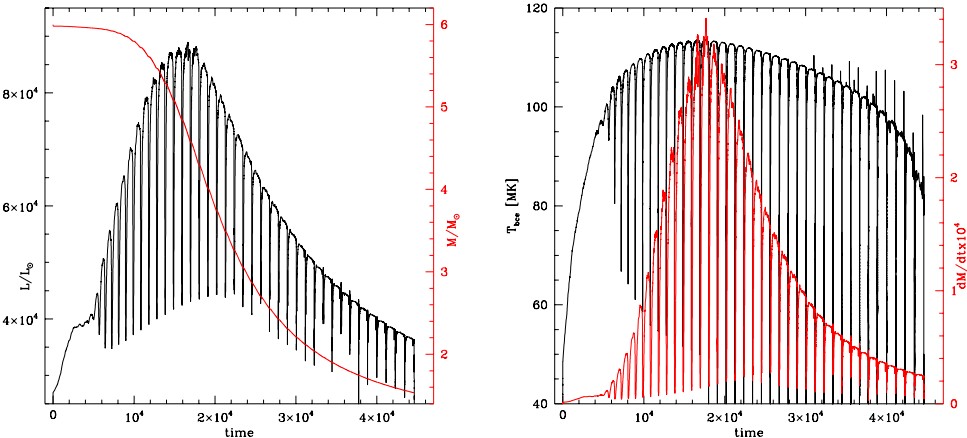

**Figure 1.** The variation with time (counted since the beginning of the AGB phase) of luminosity and current mass of the star (**left panel**, black and red tracks, respectively), and of the temperature at the bottom of the envelope and the mass-loss rate (**right panel**, black and red lines, respectively.)

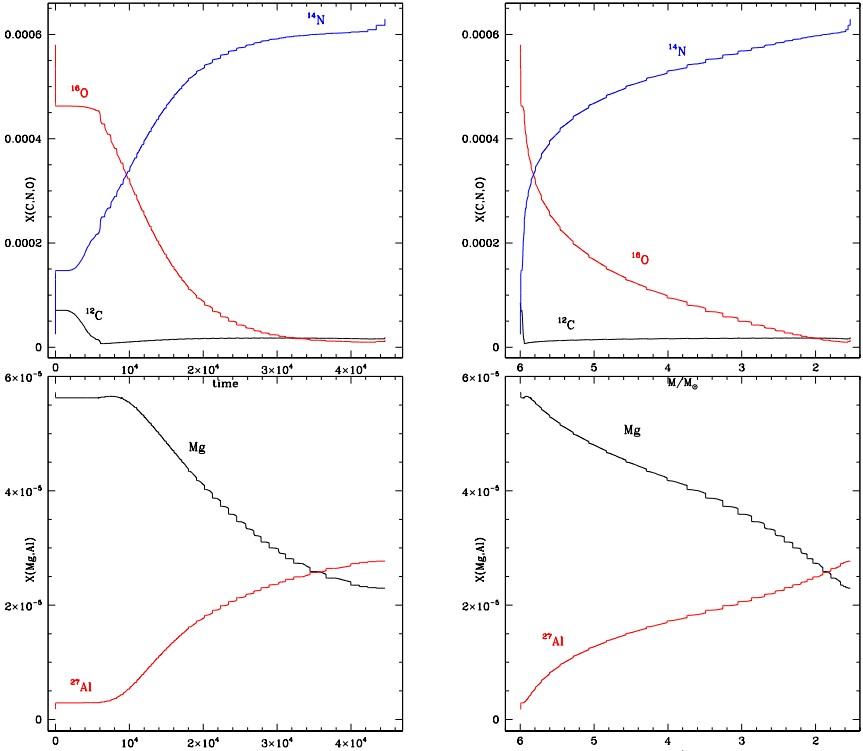

**Figure 2.** The variation of the surface chemical composition of a 6 $M_\odot$ model star of metallicity $Z = 10^{-3}$ in terms of the abundances of the most stable isotopes taking part to the CNO nucleosynthesis (**upper panels**) and of magnesium and aluminium (**bottom panels**). The various quantities are shown in terms of the AGB lifetime (**left panels**) and of the current mass of the star (**right panels**).

On the chemical side, the ignition of HBB alters the surface abundances of the elements involved in the CNO nucleosynthesis. As shown in the top, left panel of Figure 2, as soon as HBB is activated the surface $^{12}$C is depleted by one order of magnitude, whereas $^{14}$N increases by a factor $\sim 10$; this is the clue that the CN cycling is activated. After $\sim 7$ Kyr

the temperatures in the inner regions of the envelope reach ∼80 MK, thus triggering the ignition of the full CNO cycle [34,40]. This is evident in the decrease in the surface $^{16}$O. We note that in this second phase the surface mass fraction of $^{12}$C slightly increases, as a consequence of the depletion of the surface $^{16}$O. Overall, in the 6 M$_\odot$ case shown in Figures 1 and 2, the surface mass fractions of $^{12}$C and $^{16}$O are decreased by a factor ∼5 and ∼40, respectively, during the whole AGB phase, while $^{14}$N is enhanced by a factor ∼20 [1].

While the energy release by AGB stars is mostly due to the reactions taking part to CNO cycling, the ignition of HBB also affects the surface chemical abundances of other species exposed to proton-capture nucleosynthesis. An example is the synthesis of aluminium via proton capture by magnesium nuclei: the bottom, left panel of Figure 2 shows the evolution of the surface mass fractions of Mg and Al, where the depletion of the overall Mg (by a factor ∼2) and the increase in the Al content (a factor ∼15) of the envelope are evident.

The run of the surface chemistry as a function of the mass of the star, reported in the right panels of Figure 2, gives an indication of the average chemical composition of the gas ejected from these stars. An important information coming from the analysis of the left panels of Figure 2 is that the possibility that the ejecta are deeply altered with respect to the initial chemistry, and that they are significantly enhanced in nitrogen and aluminium, and depleted in oxygen and magnesium, is determined by the mass-loss rate experienced by the stars during the initial AGB phases: if little mass is lost during the phases following the ignition of HBB, the surface chemistry is changed significantly while the mass of the star is almost constant, thus most of the mass is lost after the surface chemistry was altered by HBB, and the chemistry of the ejecta will be more extreme. Among the various chemical species shown in Figure 2, carbon is the least affected by the mass-loss rate experienced by the stars across the AGB phase: this is in agreement with the discussion in Section 3.2 (see also top panels of Figure 2), that the surface carbon keeps almost constant during the AGB phase, particularly after the whole CNO cycling is activated.

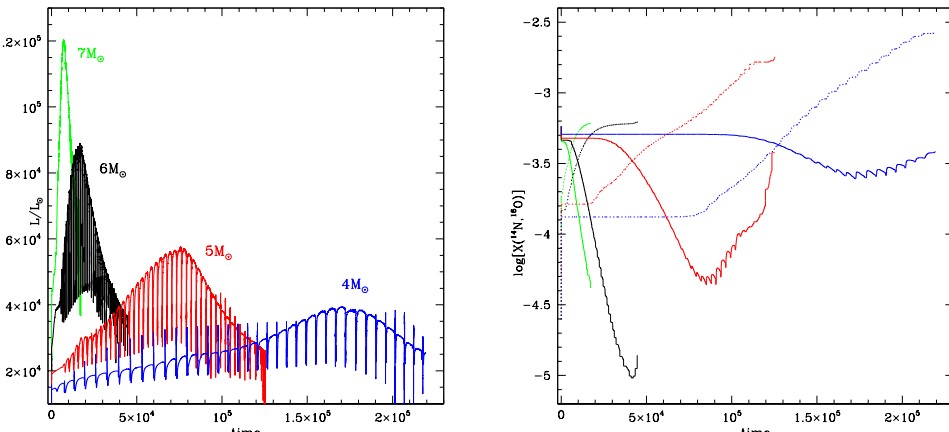

**Figure 3.** Time variation of the luminosity (**left panel**) and of the surface chemistry (**right**) of model stars of metallicity $Z = 10^{-3}$ and initial mass in the 4–7 M$_\odot$ range. The surface mass fractions of $^{16}$O (solid tracks) and $^{14}$N (dotted) are indicated in the right panel.

### 3.3. The Role of the Stellar Mass

While all the stars of mass above ∼4 M$_\odot$ are exposed to HBB, the strength of this phenomenon, and more generally the evolution of the main physical quantities characterizing a given star, are sensitive to the initial mass. This is because the higher the initial mass, the bigger and more massive the degenerate core during the AGB phase, the brighter the star.

This general behaviour with the initial mass of the stars can be seen in the left panel of Figure 3, which shows the time variation of the luminosity of model stars of metallicity $Z = 10^{-3}$, and initial masses 4, 5, 6 and 7 M$_\odot$. We note the significant differences in the luminosities reached by the various model stars, with peak values of $4 \times 10^4$ L$_\odot$ (4 M$_\odot$), $6 \times 10^4$ L$_\odot$ (5 M$_\odot$), $9 \times 10^4$ L$_\odot$ (6 M$_\odot$), $1.2 \times 10^5$ L$_\odot$ (7 M$_\odot$). The differences in the luminosity

also affect the duration of the AGB phase, which for this particular metallicity ranges from 20 Kyr, for the 7 $M_\odot$ model star, to 220 Kyr, for M = 4 $M_\odot$.

The variation of the surface chemical composition of the stars is shown in the right panel of Figure 3, in terms of the variation of $^{14}$N and $^{16}$O. The surface $^{16}$O decreases during the AGB phase, owing to the action of proton-capture reactions at the base of the envelope. This general trend is partly reversed during the final AGB phases, after HBB is practically turned off, and the surface oxygen increases under the effects of the third dredge-up (TDU) mechanism [41], which brings to the surface regions of the stars with the products of the helium-burning nucleosynthesis, primarily carbon and oxygen, which is activated in the convective shell which forms at each TP [32].

From the results reported in the right panel of Figure 3 we note that the extent of the depletion of the surface $^{16}$O increases with the initial mass of the star, in the M $\leq$ 6 $M_\odot$ domain: this is because the temperature at the base of the envelope, thus the strength of the HBB experienced, increases with the mass of the star. This trend is reversed for M > 6 $M_\odot$, because the mass-loss rate is so large that the envelope is consumed before a very advanced nucleosynthesis is activated [37]. The trend of the surface $^{14}$N also turns out to depend on the initial mass of the star, although the trend with mass is more tricky than in the oxygen case. The reason is that in the stars of initial mass in the 4–5 $M_\odot$ range the synthesis of nitrogen is not only due to the carbon and oxygen initially present at the base of the envelope, but also to the primary carbon and oxygen transported to the surface regions via TDU. The surface $^{12}$C does not show up the same sensitivity to the mass of the star as in the $^{16}$O case: the reason is once more connected with the fact that the surface carbon soon reaches the equilibrium value, and does not change significantly afterwards.

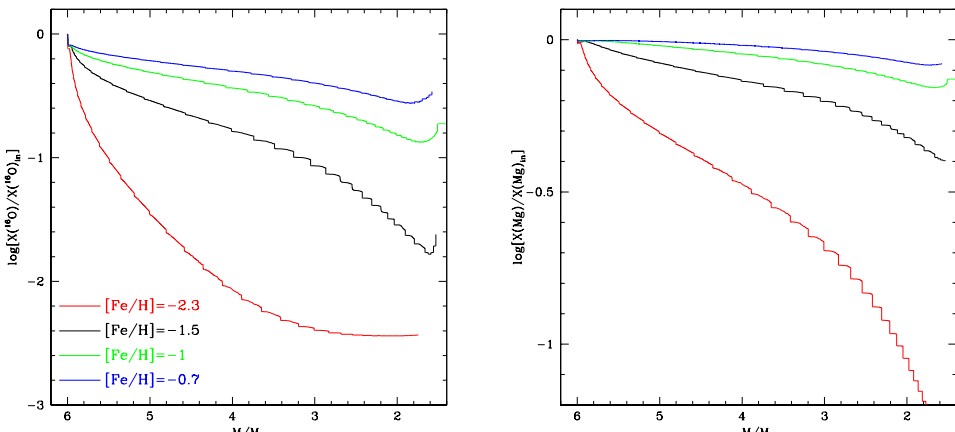

**Figure 4.** Variation of the surface mass fraction of $^{16}$O (**left panel**) and magnesium (**right**) as a function of the current mass of model stars of initial mass 6 $M_\odot$ and different metallicities. The mass fractions reported on the *y*-axis are in a logarythmic scale, and divided by the initial abundance.

### 3.4. The Effects of Metallicity

The initial mass of the star is not the only factor affecting the strength of the HBB experienced by massive AGBs, and the corresponding alteration of the surface chemistry. The detailed analysis by [34] outlined the important role played by the metallicity of the star on regard of the thermodynamic stratification of the external regions, particularly the temperature at the bottom of the envelope. [34] demonstrated that while the CN cycle is activated by all massive AGBs, the ignition of further proton-capture channels, particularly those leading to the depletion of the surface oxygen and magnesium, are extremely sensitive to the metallicity: the lower the metallicity of the star, the hotter the base of the convective envelope, the higher the strength of the HBB activated.

Figure 4 shows the variation of the surface mass fraction of oxygen and magnesium during the AGB lifetime of model stars of initial mass 6 $M_\odot$ and different metallicities, which correspond to metal-poor, intermediate metallicity and metal-rich GCs. The choice

of the initial mass is based on the results discussed in Section 3.3, which showed that 6 $M_\odot$ stars are those experiencing the most advanced nucleosynthesis, thus their chemistry is a reliable indicator of the largest variation of the chemical composition of the ejecta with respect to the initial chemical composition that we should expect.

The results reported in the left panel of Figure 4 outline the notable sensitivity of the oxygen depletion $\delta(^{16}O)$ to the metallicity. In the lowest metallicity case, i.e., $[Fe/H] = -2.3$ [2], typical of the most metal-poor GCs in the Milky Way, such as M 15 and M 92, the depletion factor is $\sim 250$. The extent of $\delta(^{16}O)$ decreases with metallicity, being only a factor $\sim 3$ in the most metal rich chemistry shown on the figure, i.e., $[Fe/H] = -0.7$, typical of metal-rich clusters, such as 47 Tuc.

The depletion factor of magnesium, $\delta(Mg)$, shown in the right panel of Figure 4, is also sensitive to the metallicity. However, in this case the slope of the $\delta(Mg)$ vs $[Fe/H]$ relationship is less steep than oxygen, and furthermore $\delta(Mg)$ tends to vanish in the $[Fe/H] > -1$ domain: therefore the use of magnesium to study the strength of the HBB nucleosynthesis by massive AGBs must be limited to the more metal-poor chemistries.

From the results shown in Figure 4, combined with the earlier discussions on the evolution of the surface carbon in Sections 3.2 and 3.3, we deduce the oxygen, among the chemical species that are exposed to proton-capture by HBB, is the one whose variation during the AGB phase is most sensitive to the metallicity of the stars considered.

## 4. The Self-Enrichment Scenario by AGBs

The idea that some self-enrichment mechanism operated in Galactic GC, and favoured the formation of multiple populations, arose when it became evident that the chemical patterns traced by stars in GC are not detected in field dwarfs of the Milky Way. This can be only explained by considering that each GC forms and is initially populated by a first generation of stars (1G), and that in later epochs a further, second generation of stars (2G) forms, from the ashes of 1G stars. [40] proposed that 2G stars formed from the gas ejected by massive AGBs, which evolve on time scales shorter than 200 Myr.

### 4.1. The Interpretation of the Chemical Patterns of GC Stars Based on the Chemistry of the AGB Ejecta

Understanding whether the chemical patterns traced by stars in a given GC can be explained within the framework of the self-enrichment scenario by massive AGBs requires the comparison between the chemical composition of the stars, in particular the mass fractions of the individual species, with the chemistry of the ejecta of models of massive AGBs (descending from $M \geq 4\,M_\odot$ progenitors) with the same chemical metallicity of the cluster stars. On the graphical side, this is commonly done by considering two chemical species ($i$ and $j$) per time, and representing the chemical composition in the $[i/Fe]$ vs $[j/Fe]$ plane, which are connected to the mass fractions $X^i$ and $X^j$.

The starting point of this kind of analysis is the comparison between the chemical composition of the stars in the cluster exhibiting the most anomalous chemistry (usually those with the lowest oxygen and magnesium) with the ejecta of $\sim 6\,M_\odot$ stars, which are generally those characterized by the largest deviations with respect to the initial chemical composition, in agreement with the discussion in Section 3.3.

However, the plain comparison between the AGB ejecta and the observations is not the full story, as the observational results obtained so far have revealed clear (continuous) chemical patterns, that can be interpreted only by assuming dilution between the winds from AGBs and pristine gas in the cluster, the latter characterized by the same chemistry of 1G stars. In this context, the clusters stars exhibiting the most extreme chemical composition are those formed from the AGB ejecta, with little or no dilution with pristine gas. The possibility that pristine survives to the epoch of type II SNe explosions was investigated by [42], where the conditions under which re-accretion of pristine gas to the cluster after the type II SNe explosion epoch are extensively discusses.

The need of considering dilution pushed researchers to use dilution curves on the $[i/\text{Fe}]$ vs $[j/\text{Fe}]$ planes introduced earlier in this section, built by mixing pristine gas, characterized by mass fractions $X_0^{i,j}$ (where $i,j$ indicates the i-th and j-th gas species), with the AGB ejecta, with abundances $X_{ej}^{i,j}$. The parameter changing across each dilution curve is the fraction of pristine gas, $f$. Therefore, the parametric equation of each curve is given in the form:

$$X^{i,j} = f \times X_0^{i,j} + (1 - f) \times X_{ej}^{i,j}$$

The final step consists in the overlapping of the dilution curves with the abundances of the individual stars, to check consistency between the observations and the theoretical expectations, and to deduce the fraction of gas pristine (if any) absorbed during the dilution process.

### 4.2. The Reconstruction of the Star Formation History in NGC 2808

The idea of pollution from massive AGBs with pristine gas in the cluster was used by [16] to simulate, for the first time on the basis of a hydrodynamical approach, the evolution of the cluster NGC 2808 [3]. This hypothesis was shown to work in the case of NGC 2808, for which it was suggested that $\sim$20% of the stars formed $\sim$40–50 Myr after the formation of the cluster, directly from the ejecta of the most massive AGBs, whereas $\sim$30% of the stellar population formed in a later epoch, from matter lost by AGBs mixed with pristine gas. This distribution was shown to be tightly correlated with the morphology of the HB and the distribution of the stars along the main sequence: (a) 1G stars populate the red side of the HB and of the MS; (b) the stars formed directly from the ejecta of the AGB, with helium Y$\sim$0.35 ([33], see also Section 3.1), evolve on the bluest and faintest side of the HB, and lay along the blue side of the MS; (c) the stars formed from dilution of AGB ejecta and pristine gas are distributed on the colour-magnitude diagram between those belonging to groups (a) and (b).

### 4.3. The Self-Enrichment Scenario Applied to the Clusters Observed by APOGEE

In a more recent exploration [34] presented a detailed analysis of the chemical patterns exhibited by the stars belonging to Galactic GCs of different metallicity, with the goal of checking consistency between the extent of the spread of the abundances of the different chemical species and the predictions from the self-enrichment by AGB scenario. The analysis by [34] was based on the comparison between the chemical composition of the stars exhibiting the most anomalous chemistry and the theoretical ejecta from massive AGB stars with the same metallicity of the clusters investigated. The chemical species taken into account by [34] were oxygen, magnesium and aluminium, for which accurate abundances estimates by the Apache Point Observatory Galactic Evolution Experiment (APOGEE, [44]) are available.

The main conclusion reached by [34] was that the chemistry of the AGB stars exhibiting the largest deviation from the initial chemical composition of the gas from which they formed, generally those of initial mass $\sim 6\,M_\odot$, are in nice agreement with the chemistry of the most extreme stars; this conclusion held for all the GC discussed.

### 4.4. The Information Deduced from the Analysis of the HB Morphology

In the present investigation we propose a more accurate analysis than the one presented by [34], as we also consider the degree of dilution with pristine gas, when comparing the yields from AGBs with the observations. Indeed the chemistry of the most extreme stars in a given GC reflects the pure chemical composition of the ejecta only in the case that the formation of the 2G took place directly from the winds of AGBs, with no dilution with pristine gas. The plain comparison between the chemistry of the ejecta and that of the stars exhibiting the most extreme chemical composition can provide only a rough indication of

the possibility that self-enrichment by massive AGBs could work in a specific GC, but it cannot provide detailed information on the modality of the star formation of 2G stars.

The degree of dilution cannot be directly derived from the observations. On the other hand a robust indication of the degree of dilution is obtained by the distribution of the helium abundances of stars in a given cluster, which can be obtained by the morphology of the HB and/or by the width of the main sequence. The key point here, discussed in Section 3.1, is that the ejecta of massive AGBs are enriched in helium, and that the largest helium in the ejecta is Y~0.35, this quantity being fairly independent of the metallicity of the cluster. Therefore, the plain comparison between the chemistry of the stars in a cluster and the ejecta from AGBs is reasonable only if there are indications that the same cluster harbours a stellar population with helium Y~0.35, as it is the case for NGC 2808 [9,13]. Should the largest helium of the stars in the cluster be smaller, dilution of the ejecta with pristine cluster must be invoked, thus the comparison must be done only after the effects of dilution are considered, which means that the ejecta of AGBs must be mixed with gas sharing the same chemistry of the 1G of a given cluster.

The helium spread among stars are nowadays available for almost the totality of the Galactic GCs, thanks to the UV Legacy Survey presented by the research team led by Prof. Milone in a series of papers [11,20,45]. This opens the possibility of a deeper analysis of the reliability of the self-enrichment by massive AGB scenario for the various GC in the Milky Way.

In the next section we apply this method to interpret the stellar populations nowadays evolving in the cluster NGC 6402. This is the standard method which will be applied in the future to Galactic GCs.

## 5. Understanding Star Formation in NGC 6402

In a recent work [46] analyzed the stellar populations of the cluster NGC 6402. Results from high-resolution spectroscopy, obtained by [47], outline clear trends involving the abundances of magnesium, aluminium and oxygen of the stars observed. The detailed analysis of the morphology of the HB of this cluster in the UV bands suggests that the cluster harbours two distinct populations, separated by a helium spread $\delta Y = 0.06$–$0.07$: therefore, assuming that the helium of the 1G is Y = 0.25, we deduce that the population of the cluster with the most extreme chemistry is characterized by a helium mass fraction not higher than Y= 0.32. If we consider that the helium content of the ejecta of massive AGBs is Y~0.35, the upper limit on the helium content of the stars populating the HB indicates that during the formation of the 2G the gas from AGBs was diluted with at least 30% of pristine gas when the 2G of the cluster formed. This is the quantity we must take into account when comparing the chemistry of the ejecta of AGBs with the chemical patterns traced by the cluster stars.

### 5.1. Yields from Massive AGB Stars with the Same Chemistry of NGC 6402

We start by following the same approach as in [34], and compare in Figure 5 the yields from the model stars obtained with the standard input described in Section 2 with the results from the observations, taken from [47]. We note that the evolutionary sequences for this analysis have been calculated on purpose, by assuming the same chemical composition (we refer in particular to the metallicity and the $\alpha$ enhancement) of the 1G of the cluster, as given in [47].

The results obtained by diluting the ejecta from the model stars [4] with various percentages of pristine gas sharing the same chemistry as the 1G of the cluster are indicated with red circles in Figure 5. The chemistry of the yields from AGBs (represented by the points on the left side of the planes in both panels of the figure) is fairly consistent with the average mass fraction of the most extreme stars of the 2G: the oxygen and aluminium exhibit differences of $\delta[\text{O}] = -0.7$ and $\delta[\text{Al}] = +1.2$, respectively, between 1G and 2G. The magnesium of the ejecta is a bit higher than the quantities measured in some of the 2G stars,

although the analysis is more tricky in this case, as the differences are comparable with the errors associated to the individual measurements, which are of the order of 0.1 dex [47].

Following the discussion done in the previous section, in the present case we cannot test the hypothesis that the 2G formed from the ashes of massive AGBs by the plain comparison of the ejecta with the observations, as we must consider that at least 30% of pristine gas took part into the formation of the 2G. Therefore, the chemical composition of the 2G stars of NGC 6402 with the most extreme chemistry must be compared with the point along the dilution sequence corresponding to a fraction of 70%(30%) of the AGB gas (pristine gas). Under this perspective the comparison cannot be regarded satisfactory, because the expected reduction of the surface oxygen is $\sim 0.3$ dex, whereas at least $\sim 0.6$ dex are required. A similar issue holds for magnesium, for which we see that the expected depletion is below 0.1 dex, whereas something around 0.15 dex is needed.

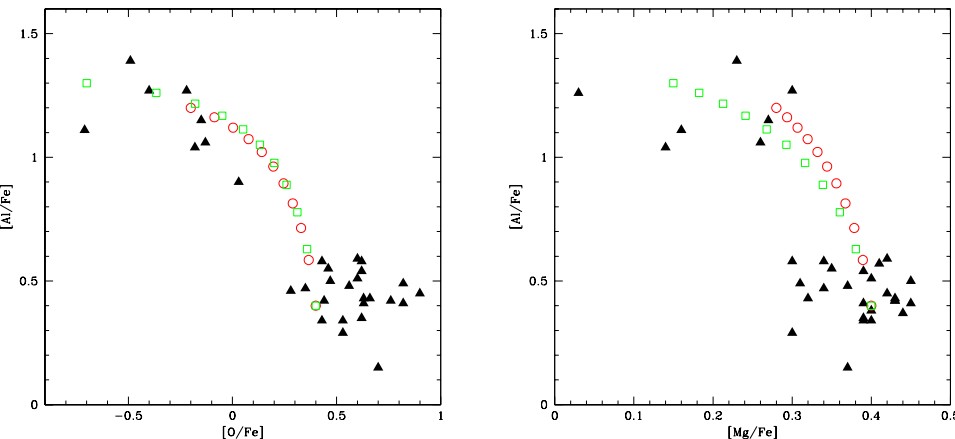

**Figure 5.** The surface chemistry of NGC 6402 stars by [47], shown in the O-Al (**left panel**) and Mg-Al (**right**) stars. Red points indicate the chemistry obtained by mixing the ejecta from AGBs with different percentages (spaced by 10%) of pristine gas, with the same chemistry of 1G stars. Green points refer to dilution of AGB ejecta obtained by assuming lower rates of mass loss than in the standard evolutionary sequences.

The results shown in Figure 5 indicate that, if the 2G of NGC 6402 formed from the gas ejected by 1G massive AGBs, the oxygen and magnesium content of the ejecta of the latter stars are too large to be consistent with results from high-resolution spectroscopy: in other words, the gas from which 2G stars formed shows up the signature of a proton capture nucleosynthesis, which is more advanced than what found by diluting the ejecta of massive AGBs with 30% pristine gas.

*5.2. Lower Mass-Loss Rates from Metal-Poor, Massive AGB Stars?*

In Section 3.2 we discussed how mass-loss affects the chemical composition of the ejecta, outlining that the difference between the chemical composition of the ejecta and the gas from which the star formed gets larger the lower the mass-loss rates experienced during the initial part of the AGB phase. To reconcile the theoretical predictions with the observational evidence, we therefore considered the possibility that the mass-loss rates during the evolution of massive AGBs are overestimated, thus we calculated new evolutionary sequences with the same chemical composition of 1G stars of NGC 6402, where the free parameter entering the Blocker's description [24] was allowed to vary in the 0.002–0.02 range.

Before describing the results, we believe important to discuss the reliability of this choice. We recall that the mass-loss treatment by [24] was calibrated on the basis of results of wind dynamics, where mass loss is eventually triggered by the effects of radiation pressure on silicate grains, which are the dominant dust species forming in the wind of M-type AGBs [48]. The results by [24] are based on hydrodynamical simulations by [49],

which in turn are based on solar metallicity environments. Because the amount of silicon in the stars scale approximately with metallicity, it is reasonable to expect that in pop II stars, such as those formed in GCs, the amount of silicate dust formed in the winds of AGBs is lower, and so the rate of the radiation driven mass-loss. The possibility that the mass-loss rates experienced by pop II massive AGBs are smaller than their solar-metallicity counterparts is therefore consistent with the mechanism of dust-driven stellar winds, at the base of the mass-loss process of this class of stars.

In Figure 6 we compare the results regarding a 6 M$_\odot$ model star with the same chemical composition of 1G stars of NGC 6402, obtained by adopting various values of the mass-loss parameter $\eta_R$. In agreement with the previous discussion, we find that the lower $\eta_R$ the higher the difference between the chemical composition of the ejecta and the initial chemistry of the star.

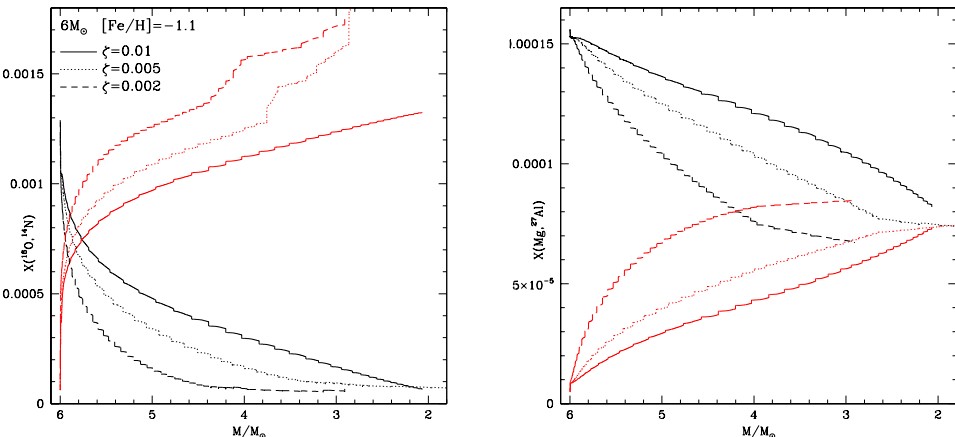

**Figure 6.** Variation of the surface chemical composition of a 6 M$_\odot$ model star with the same chemical composition of the stars belonging to the 1G of the cluster NGC 6402, as a function of the current mass of the star. The different lines refer to evolutionary sequences calculated by adopting different values of the mass-loss parameter $\eta_R$. The evolution of the surface mass fraction of $^{14}$N and $^{16}$O are shown in the **left panel**, whereas those of Mg and $^{27}$Al are reported in the **right panel**.

The application of the AGB yields obtained by adopting $\eta_R = 0.002$ to the study of NGC 6402 is reported in Figure 5, where the corresponding dilution curve is indicated with open, green squares. It is clear the improvement in the comparison between the observations and the theoretical expectations. This analysis suggests that the data of NGC 6402 stars can be explained in the context of a self-enrichment process by AGB stars, provided that the mass-loss rates experienced by AGB stars with the same chemistry of the 1G of the cluster is significantly lower than those experienced by the counterparts of solar metallicity.

## 6. The Role of Oxygen

In the previous section we showed the sensitivity of the chemical composition of the ejecta from AGB stars to the rate of mass loss experienced. Adopting a reduced mass-loss rate during the initial AGB phases makes the ejecta to exhibit a deeper signature of the proton-capture nucleosynthesis activated in the innermost regions of the envelope: the average mass fractions of oxygen and magnesium get smaller, whereas those of nitrogen and aluminium are higher. No difference is expected in the case of carbon, according to the discussion of Section 3.

The behaviour of the nitrogen and aluminium content of the ejecta of the model star are presented with red lines in Figure 6: when the mass-loss parameter changes from $\zeta = 0.02$ (the standard value) to $\zeta = 0.002$, the content of both species increases by ∼30%, which corresponds to variations in [N/Fe] and [Al/Fe] of ∼0.12 dex. There is no way to obtain largest variations, even if lower mass-loss rates were considered. For what

attains nitrogen, this is due to the fact that we are already very close to the maximum abundance that can be obtained, which corresponds to the case where the overall oxygen is processed within the CNO cycle, as is the case here. Regarding aluminium, a more advanced nucleosynthesis would make the Al equilibrium abundance to decrease, as the whole Mg-Al-Si nucleosynthesis would lead to an increase in the silicon mass fraction [50].

For what concerns magnesium, the average Mg content in the ejecta diminishes by ∼30% when the mass loss rate is decreased by a factor ∼10, which corresponds to a decrease in [Mg/Fe] of ∼0.1 dex. Unfortunately, as discussed in Section 3.4, the Mg-Al nucleosynthesis is activated efficiently only at metallicities $[\mathrm{Fe/H}] \leq -1.5$, thus no significant indications from the Mg spread exibited by the stars in the cluster can be deduced for NGC 6402.

The chemical species mostly affected by the description of mass loss is oxygen: the average oxygen in the ejecta decreases by a factor ∼3 when $\eta_R$ dimishes by a factor 10, with reflects into a 0.5 dex change in [O/Fe]. Unlike magnesium, some oxygen depletion occurs at all metallicities [34], and the sensitivity of the oxgen reduction to the strength of the HBB activated is significantly higher than magnesium.

Based on the above arguments, and considering the little help that comes from carbon on this regard, we conclude that the knowledge of the star-to-star differences in the surface oxygen abundance is the most relevant parameter to understand the nucleosynthesis at which the gas from which 2G stars in GC formed. The distribution of the oxygen abundances of the stars of a given cluster, particularly of those with the lowest oxygen, can also be used as an independent test of the degree of dilution with pristine gas that took place during the formation of the 2G.

## 7. Conclusions

We study the possibility that multiple populations in GC, whose presence has been confirmed by results from high-resolution spectroscopy and photometry, formed by the action of a self-enrichment mechanism, according to which new generation of stars were born from the ashes of massive AGB stars belonging to the original stellar population.

We test the predictions of the AGB scenario against the chemical patterns traced by RGB stars nowadays evolving in the cluster NGC 6402, particularly the O-Al and Mg-Al anti-correlations. The analysis proposed here, unlike previous studies, takes into account the effects of dilution of the AGB ejecta with pristine gas, whose extent was derived based on the width of the MS of the cluster and the morphology of the HB.

This analysis showed that consistency between theoretical predictions (i.e., the chemical composition of the ejecta from massive AGB stars) and the observational evidence is possible if the adopted mass-loss rates of massive AGBs significantly lower than those experienced by stars of similar mass and solar metallicity. This hypothesis if fully justified based on the studies of dust-driven winds of M-type AGB stars.

Among the various chemical species, oxygen proves the most reliable indicator of the degree of the nucleosynthesis at which the matter from which the stars belonging to the second generation of the clusters formed was exposed.

**Author Contributions:** Conceptualization, E.H.-M.; Investigation, C.G. and P.V.; Supervision, E.H.-M.; Writing—original draft, P.V. All authors have read and agreed to the published version of the manuscript.

**Funding:** This research received no external funding.

**Conflicts of Interest:** The authors declare no conflict of interest.

## Notes

[1]   This is the final chemistry of the star. The average chemical composition of the ejecta will show a milder variation with respect to the initial chemistry, as the average over the whole AGB lifetime should be considered.

[2]   We use the standard notation, where $[\mathrm{Fe/H}] = \log[X(\mathrm{Fe})/X(\mathrm{H})] - \log[X(\mathrm{Fe})/X(\mathrm{H})]_{\odot}$

[3]   In the work by [16] it is assumed that the mass of the 2G stars is in the 0.1–0.8 M$_{\odot}$ range. This is in agreement with previous studies by [15,43], who showed that the IMF of 2G stars must be much flatter than that of 1G, independently of the nature of the pollutants. However, even in case that massive 2G stars form, the number would be so small that their explosion would not be able to halt the further formation of the SG in the cluster.

[4]   We show the chemistry of the yields of the 6 M$_{\odot}$ model star in this plot, as this is the star from which we obtain the largest deviation from the chemistry of 1G stars.

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
