# Peer review of "Self-Enrichment in Globular Clusters: The Crucial Role Played by Oxygen"

_universe, doi:10.3390/universe9020084_

Round 1
Author Response
First we would like to thank the referee for the careful reading of the manuscript and the comments, which we considered in their entirety, which helped improve the quality of this work.
We provide below the answers to the individual points raised by the referee. Please note that the changes in the text have been outlined in bold.
R) First generation form stars with all masses, therefore there should be as well a few supernovae enriching the intra-cluster medium, do you agree? Could you comment on this.
A) We thank the referee for this comment. Indeed the works by Prantzos & Chrabonnel (2006) and by Decressin et al. (2007), now included among the references, showed that the IMF of the second generation is such that most of the stars formed are in the 0.1-0.8Msun range. This is also the hypothesis at the base of the work by D'Ercole et al. (2008), on NGC 2808. Now, the possibility that a few massive stars, that undergo core collapse, cannot be disregarded. However, this is not going to halt the formation of the second generation in the cluster, a process which required short time scales of the order of 100 Myr. Maybe the injection of positive energy might prevent star formation close to the possible explosion, but it is definitively not sufficient to halt the formation of the second generation in the whole cluster. We added a comment to the text to discuss this point raised by the referee (footnote 3, page 9, in the current version of the manuscript)
2) Your results need remaining pristine gas, to be mixed with the AGB ejecta, but could it not have been all ejected from the cluster by the supernovae winds ?
We understand the referee's point here. Definitively most of the pristine gas is blown away during the type II supernovae explosion. However, the detailed work by D'Ercole et al. (2016), now added to the references, showed that in the context of the self-enrichment by AGBs scenario part of the pristine gas is pushed away by the supernovae explosions and stored in cavities, from which it can be re-accreted to the central regions of the cluster. We added a sentence on this in page 8 (second-last paragraph).
Minor corrections:
R) Abstract, line 9: We also comment of =) We also comment on
A) changed
R) Introduction, line 5: which suggest the coexisting =) which suggest the coexistence
A) changed
R) Introduction, line 10: you cite multiple main sequences in NGC 2808 as the break-through. But this was already seen in ! Centauri by Bedin et al. (2004, ApJ, 605, L125)with HST, and previously by Lee et al. (1999, Nature, 402, 55) and Pancino et al. (2003, MNRAS, 345, 683).
A) We thank the referee for this. We added all the references suggested by the referee to the introduction
R) Introduction, 2nd paragraph, 2nd line: stars formed in GC =) stars formed within GC (just a suggestion)
A) done
R) page 2, Sect. 3, 2nd line: burning by all the star with mass =) “burning by all the stars with mass”
A) changed
R) Sect. 3.2, 4th paragraph, line 7: the loss the envelope =) the loss of the envelope
A) changed
R) Fig. 2: please write the name of the element in the curves of the right panel as well
A) done
R) Sect. 3.3, 3rd paragraph, before last line: stars the products =) stars with the products
A) changed
R) Sect. 3.3, last line: once connected =) it seems that ’once’ has no meaning here
A) we changed the sentence
R) Sect. 3.4, 3rd paragraph, line 1: results ... outlines =) results ... outline
A) changed
R) Sect. 4, 6th paragraph (p.9), line 6: quantity being fairly independently =) quantity being fairly independent
A) changed
R) Sect. 5.1, last paragraph, line 1: NGC 6492 =) NGC 6402
A) changed
Reviewer 2 Report
This study is of interest because it is related to unraveling the mystery of multiple stellar populations in globular clusters. In particular, the self-enrichment by AGBs scenario is considered. The presented analysis is more accurate than the one presented by Dell’Agli et al. (2018).
The evolution code is described. However, the description of the used methods and approaches should be a bit more detailed with a brief overview of previous approaches, results, and problems. It would be useful to give a separate section to the description of the methods.
Author Response
We thank the referee for the careful reading of the manuscript. To meet the referee's request, we split section 4 into 4 sub-sections, the first of which (section 4.1 in the current version of the manuscript) is dedicated to the explanation of the methodology followed to interpret the observations of cluster stars under the light of self-enrichment mechanisms